# Molecular Dynamics Simulations on Effect of Surface Roughness of Amorphous Substrate on Nucleation in Liquid Al

**Hua Men and Zhongyun Fan ***

Brunel Centre for Advanced Solidification Technology (BCAST), Brunel University London, Uxbridge, London UB8 3PH, Middlesex, UK

\* Correspondence: zhongyun.fan@brunel.ac.uk

**Abstract:** In this study, we used molecular dynamics (MD) simulations to investigate the atomic ordering in the liquid aluminum (Al) adjacent to the amorphous substrate with smooth and rough surfaces. This study revealed that the liquid exhibited layering within about 5 atomic layers but no visible in-plane atomic ordering at the interface with the smooth amorphous surface, and neither layering nor in-plane atomic ordering with the rough surface of the amorphous substrate. However, the smooth amorphous surface induced some local ordered structure in the liquid at the interface by a structural templating mechanism, which promoted heterogeneous nucleation by creating a 2-dimensional (2D) nucleus in the third layer. The amorphous substrate with a rough surface had no effect on the nucleation in the liquid, leading to the occurrence of homogeneous nucleation with an undercooling 100 K larger than that of heterogeneous nucleation on the smooth amorphous substrate. This study confirmed that structural templating is a general mechanism for heterogeneous nucleation.

**Keywords:** heterogeneous nucleation; homogeneous nucleation; MD simulation; interface; amorphous

## 1. Introduction

A crystalline substrate may induce pronounced atomic ordering in a liquid at the liquid/substrate interface, even at temperatures above its liquidus, and such substrate-induced atomic ordering is referred to as prenucleation [1]. The epitaxial nucleation model [2] suggests that heterogeneous nucleation proceeds layer-by-layer through a structural templating mechanism. The crystal lattice of the substrate surface provides low-energy positions for the adjacent liquid atoms, forming a locally ordered structure, which then templates the atomic ordering in the next layer. This process is referred to as structural templating. The undercooling required for epitaxial nucleation is closely related to the structural compatibility between the substrate and the solidified phase, which is often quantified by lattice misfit. Such pronounced atomic ordering can have a significant influence on the heterogeneous nucleation process [3,4]. If the atomic arrangement at the interface is compatible with the crystal structure of the new phase, it enhances heterogeneous nucleation; otherwise, incompatible atomic arrangement at the interface impedes heterogeneous nucleation. Therefore, it is important, both scientifically and technologically, to have a good understanding of how the chemical and/or physical properties of the substrate affect atomic ordering in the liquid at the interface and its implications for the heterogeneous nucleation process.

Both experimental observations [5–10] and atomistic simulations [11–16] suggest that the liquid atoms become layered within one or two nanometers at the interface (atomic layering) and that the atoms in an individual atomic layer may have certain atomic ordering (in-plane atomic ordering) at temperatures above the liquidus. The atomic ordering in the liquid at the interface can be manipulated by changing the structure and/or chemistry of the substrate. Layering has been attributed to the "hard wall" effect of the substrate surface [17]. The degree of layering is usually independent of the crystal structure [12] and

surface orientation [11,12] of a substrate with a smooth surface, and of the lattice misfit between the substrate and solid phase corresponding to the liquid [14]. All these studies suggest that atomic layering at the interface is hardly altered by changing the substrates, as long as the substrate surface is smooth at atomic scale. The in-plane atomic ordering at the interface has been attributed to the low-energy atomic positions provided by the crystalline lattice in the surface of the substrate. Therefore, the in-plane atomic ordering is closely related to the crystal structure of the substrate [11–14]. Using the molecular dynamics (MD) simulations, it was found that the in-plane atomic ordering persists within the first three atomic layers adjacent to an interface with a small lattice misfit, and becomes very weak in substrates with a large lattice misfit [14]. This suggests that the in-plane atomic ordering can be manipulated by changing the crystallographic matching between the substrate and solid upon solidification.

Prenucleation can be partially or completely demolished by impeding the "hard wall" effect and/or structural templating with a rough substrate surface [17]. Geysermans et al. [11] found that both atomic layering and in-plane atomic ordering can be suppressed by the rough surface of a bulk amorphous substrate. Galea et al. [18] investigated the effect of atomic level roughness of crystalline substrates on slip length at the fluid/solid boundary during shear flow, by varying the size and spacing of substrate atoms at a constant packing fraction. They found that the amplitude of the density oscillations at the interface increases by increasing the smoothness of the surfaces. Our previous study revealed that for a rough surface of a crystalline substrate, the "hard wall" effect [19,20] for the atomic layering is impeded while the structural templating for the in-plane atomic ordering remains to a certain degree, because every fraction of the surface layer of the substrate still provides some low-energy atomic positions in the liquid. The atomic layering and in-plane atomic ordering decrease with increasing surface roughness of a crystalline substrate [17]. For the rough surface of an amorphous substrate, the structural templating for the in-plane atomic ordering is almost completely impeded due to the disordered structure of the substrate, and the "hard wall" effect for the atomic layering is gradually impeded with increasing surface roughness of the substrate. Thus, the rough surface of an amorphous substrate almost completely eliminates the in-plane atomic ordering in the liquid regardless of surface roughness, and reduces or eliminates the atomic layering depending on the surface roughness [17].

However, it is not clear whether heterogeneous nucleation can proceed in a liquid on the surface of an amorphous substrate, because the amorphous substrate has a disordered structure. It is generally thought that the probability of heterogeneous nucleation is extremely low with an amorphous substrate. For example, it is claimed that homogeneous nucleation occurs in the droplet of liquid metals fluxed and enclosed with $B_2O_3$, as long as the droplet is small enough [21]. The objective of this study was to investigate the effect of the surface roughness of an amorphous substrate on prenucleation and nucleation.

## 2. Simulation Approach

The embedded atom method (EAM) potential for aluminum, developed by Zope and Mishin to model interatomic interactions [22], was used in this work. The predicted melting temperature for pure Al is $870 \pm 4$ K with this potential [22]. The liquid Al was prepared by heating the system to 1400 K with a temperature step of 50 K, equilibrating for 100 ps. The liquid Al was then cooled to 900 K with a temperature step of 50 K and equilibrating for 1000 ps. The amorphous substrate (3 atomic layers thick) with a rough surface was obtained by freezing the atomic positions in the bulk liquid Al equilibrated at 900 K, which is denoted as 3D amorphous substrate. The amorphous substrate with a smooth surface was obtained by keeping the and *y*- coordinates and setting the *z* coordinates to 0 for each atom in a layer of liquid Al equilibrated at 900 K with a thickness of one atomic plane spacing along the *z*-direction, and this is denoted as 2D amorphous substrate. To be consistent with the case for the 3D amorphous substrate, in the case of the 2D amorphous substrate, we used 3 layers of such 2D amorphous plane as the substrate that had the same layer spacing as

in the 3D amorphous substrate. The simulation system included 77,760 atoms in total and 6342 atoms in the substrate, with a size of 133.6 Å ($x$) × 123.4 Å ($y$) × 87.2 Å ($z$) while equilibrating at $T$ = 900 K.

During MD simulation, the atoms in the substrate were fixed, and the liquid Al atoms above the substrate were allowed to move freely under the effect of the interatomic potential. The substrate atoms were excluded from the equations of motion, but the forces they exerted on the adjacent atoms were included. All the MD simulations were performed using the LAMMPS package (12 Dec 2018 version, Sandia National Laboratories, Sandia, NM, USA) [23]. The equations of motion were integrated by means of the Verlet algorithm with a time step of 0.001 ps, and the Nose-Hoover NVT ensemble was used for the temperature control.

The nucleation temperature, $T_n$, for each specified nucleation system was determined using the variable step search method. The equilibrated configuration of the systems at 900 K was cooled to a desired temperature with a step of 50 K, and at each temperature step, the system was allowed to run for 1000 ps, which had been checked to be sufficient to reach equilibrium in the current study. The initial nucleation temperature, $T_1$, was determined by monitoring the variation in total energy (potential energy and kinetic energy of all the atoms) and trajectory of the system during the equilibration. This means that nucleation exactly occurred in the temperature interval between $T_1$ and $T_1$ + 50 K. A more accurate nucleation temperature, $T_2$, was determined by a finer search in this reduced temperature interval with a temperature step of 5 K. Finally, the nucleation temperature, $T_n$, was determined by an even finer search between $T_2$ and $T_2$ + 5 K with a temperature step of 1 K. This approach allowed the nucleation temperature to be determined within an error of ±1 K.

The atomic ordering in the liquid adjacent to the liquid/substrate interface was quantified by the atomic density profile, $\rho(z)$, for ordering along the $z$-direction, which is defined as [12]:

$$\rho(z) = \frac{< N_z >}{L_x L_y \Delta z},\tag{1}$$

where $N_z$ is the number of atoms between $z - \Delta z/2$ and $z + \Delta z/2$ at time $t$; $\Delta z$ is the width of the bin, a 10th of the layer spacing in this study. The angled brackets indicate a time-averaged quantity, and $L_x$ and $L_y$ are the $x$ and $y$ dimensions of the cell, respectively.

The atomic arrangement in the liquid adjacent to the interface during the simulation was characterized by the time-averaged atomic positions [24] and local bond-order analysis [25]. The time-averaged atomic positions in the individual layers of the liquid within 10 ps were taken from the trajectory of the simulation. With this approach, the solid atoms could be distinguished from the liquid atoms, where the solid atoms usually vibrate at their equilibrium positions while the liquid atoms can move beyond one atomic spacing [24]. The local bond-order analysis is another approach widely used in atomistic simulations to distinguish the solid from the liquid atoms in the bulk liquid [26]. To perform the local bond-order analysis, the local bond-order parameters, $q_l(i)$, were calculated as [24]:

$$q_l(i) = \left( \frac{4\pi}{2l+1} \sum_{m=-l}^{l} |q_{lm}(i)|^2 \right)^{\frac{1}{2}},\tag{2}$$

where the $(2l + 1)$ dimensional complex vector $q_{lm}(i)$ is the sum of spherical harmonics, $Y_{lm}(r_{ij})$, over all the nearest neighboring atoms of the atom $i$. Two neighboring atoms, $i$ and $j$, can be recognized as connected if the correlation function, $q_6(i) \cdot q_6(j)$, of the vector $q_6$ of neighboring atoms $i$ and $j$ exceed a certain threshold, 0.5 in this study. To distinguish the solid from the liquid atoms, a threshold on the number of connections that an atom had with its neighbors was set to 6.

## 3. Results

### 3.1. Atomic Ordering at Liquid/Substrate Interface

Figure 1a shows the front view of a snapshot at $t$ = 1000 ps during the simulation for the simulation system with a smooth-surfaced f amorphous substrate (2D amorphous, hereafter) equilibrated at 900 K. The liquid at the interface had a layered structure within a few atomic layers away from the interface. The corresponding density profile, $\rho(z)$, of the system is plotted as a function of distance, $z$, from the interface in Figure 2a. The amorphous substrate with a smooth surface had sharp peaks, as expected. The layering in the liquid at the interface persisted within about five atomic layers, and the peak density showed exponential decay. The first peak had a density of 0.13 Å$^{-3}$, which is significantly higher than 0.05 Å$^{-3}$, the average atomic density of the bulk liquid.

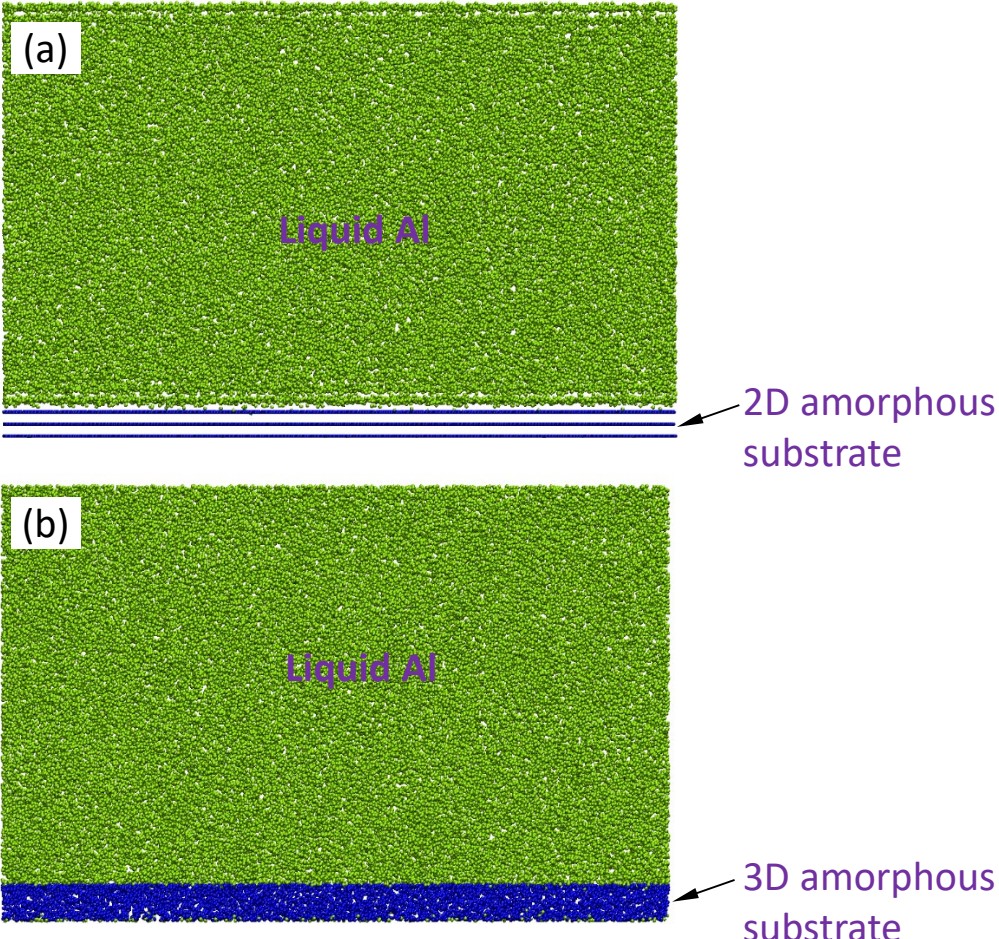

**Figure 1.** The snapshots of the simulation system of the liquid Al/amorphous substrates with (**a**) smooth surface (2D amorphous) and (**b**) rough surface (3D amorphous) at $t$ = 1000 ps during the simulation at $T$ = 900 K. The amorphous substrates with smooth or rough surfaces are obtained by quenching the liquid equilibrated at $T$ = 900 K. The liquid at the interface with 2D amorphous substrate exhibited a layered structure. In contrast, there is no layered structure in the liquid at the interface with 3D amorphous substrate.

Figure 1b shows the front-view of a snapshot of the system of amorphous substrate with an atomically rough surface (i.e., a 3D amorphous substrate) equilibrated at $T$ = 900 K at $t$ = 1000 ps. The corresponding $\rho(z)$ is plotted as a function of distance from the interface in Figure 2b. The liquid at the liquid/substrate interface did not exhibit any layering or in-plane atomic ordering (Figures 1b and 2b). Therefore, the 3D amorphous substrate could not induce any atomic ordering in the liquid at the interface.

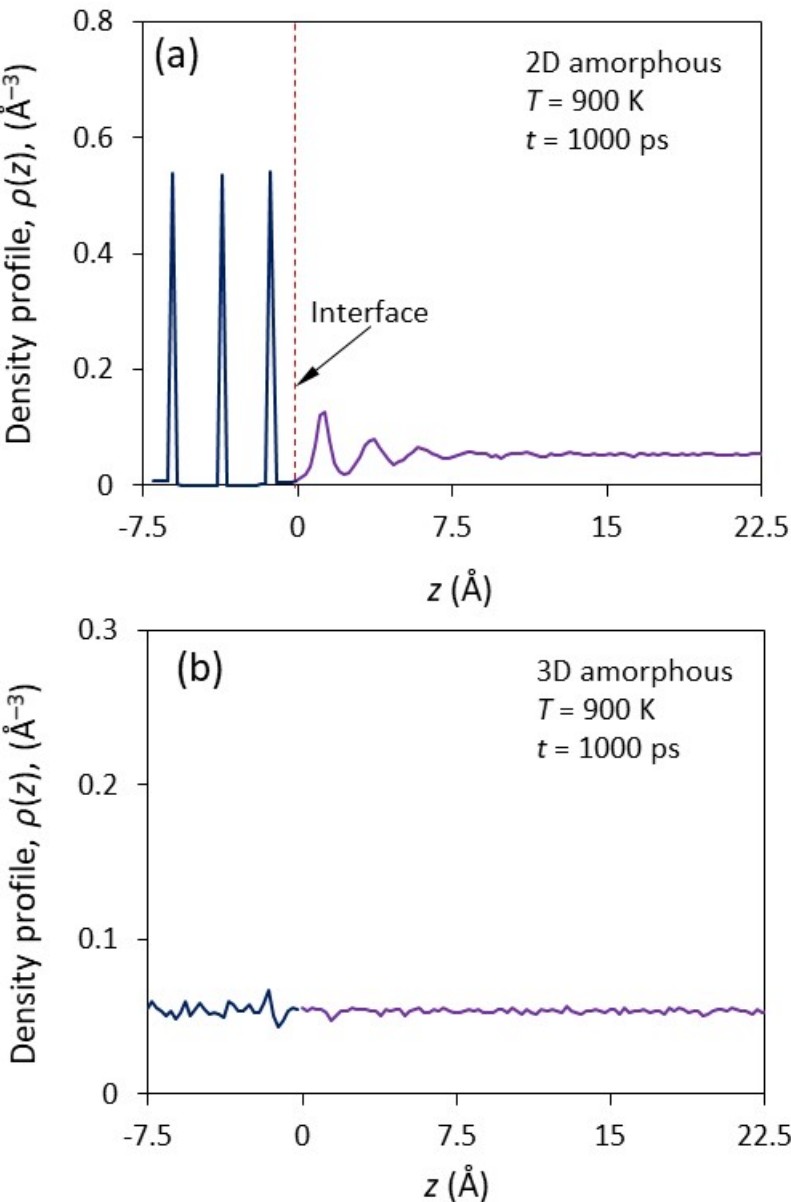

**Figure 2.** The density profiles, $\rho(z)$, as a function of the distance, $z$, away from the interface for the simulation system with (**a**) 2D and (**b**) 3D amorphous substrates at $t = 1000$ ps during the simulation at $T = 900$ K. The 2D amorphous substrate exhibits sharp peaks in the density profile. Atomic layering in the liquid at the interface with smooth surface persist within about 5 atomic layers, with its peak densities exponentially decaying. However, layering is invisible in the liquid at the interface with a rough surface.

Figure 3a–c show the radial distribution function (RDF) as a function of the radial distance, $r$, in the first interfacial layer (A1) of the substrate, and the first (L1) and second (L2) interfacial layers of the liquid at 1000 ps for the system with a 2D amorphous substrate equilibrated at $T = 900$ K. All the RDFs exhibited a first sharp peak and a second diffuse peak, suggesting that they all had a disordered structure, typical for amorphous or liquid phases. The A1, L1, and L2 layers of the system with a 3D amorphous substrate displayed almost identical RDFs to those of the 2D amorphous (Figure 4a–c). This suggested that there only existed short-range order in the amorphous substrates and the liquid at the interface, regardless of surface of the amorphous substrates being smooth or rough.

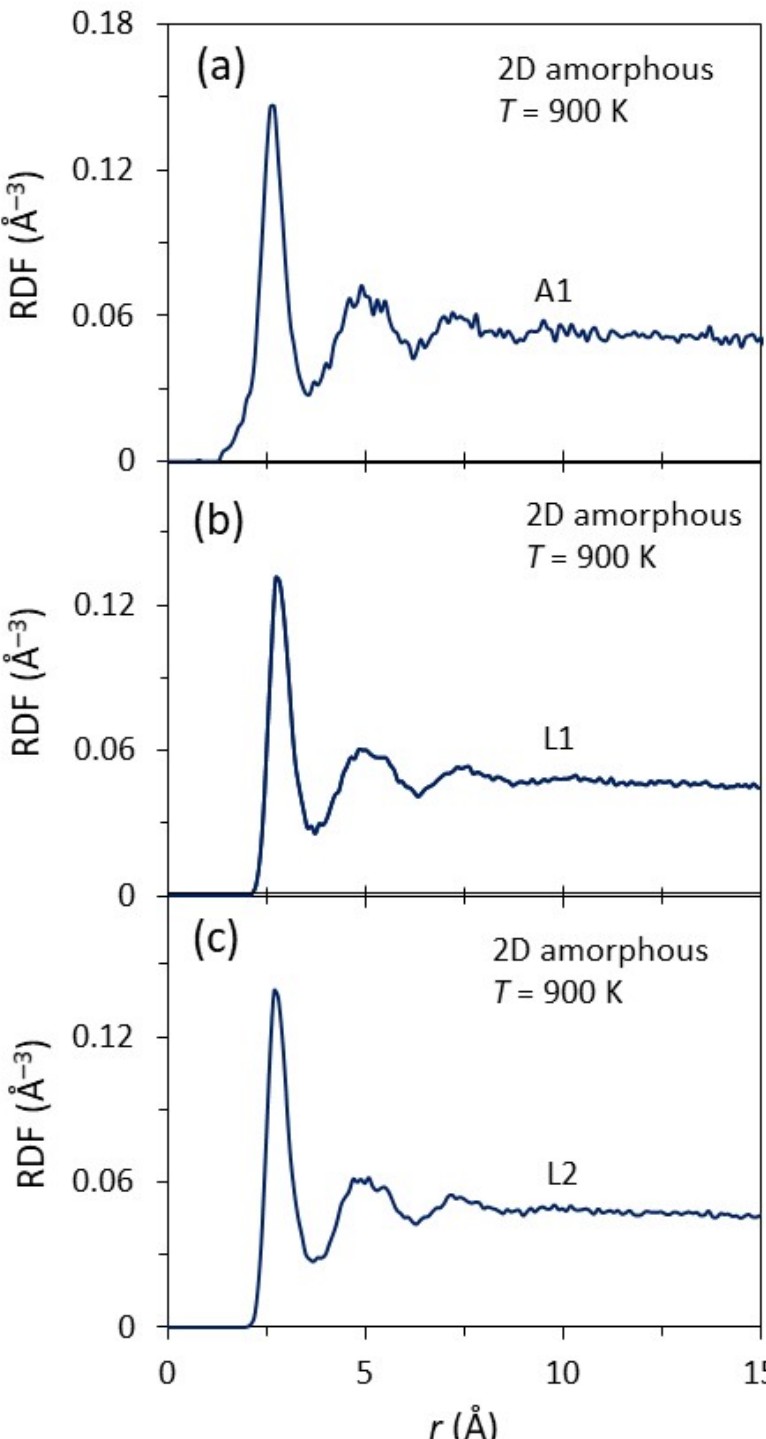

**Figure 3.** The radial distribution functions (RDFs) as a function of the distance, *r*, for (**a**) the 1st layer of the amorphous substrate (A1), and (**b**) the 1st (L1) and (**c**) 2nd (L2) layers of liquid at the interface with the 2D amorphous substrate equilibrated at *T* = 900 K. The 1st peak is sharp and the 2nd peak is diffuse in the RDFs for all the A1, L1 and L2, indicating that there is only short-range order in the amorphous substrate and the liquid at the interface.

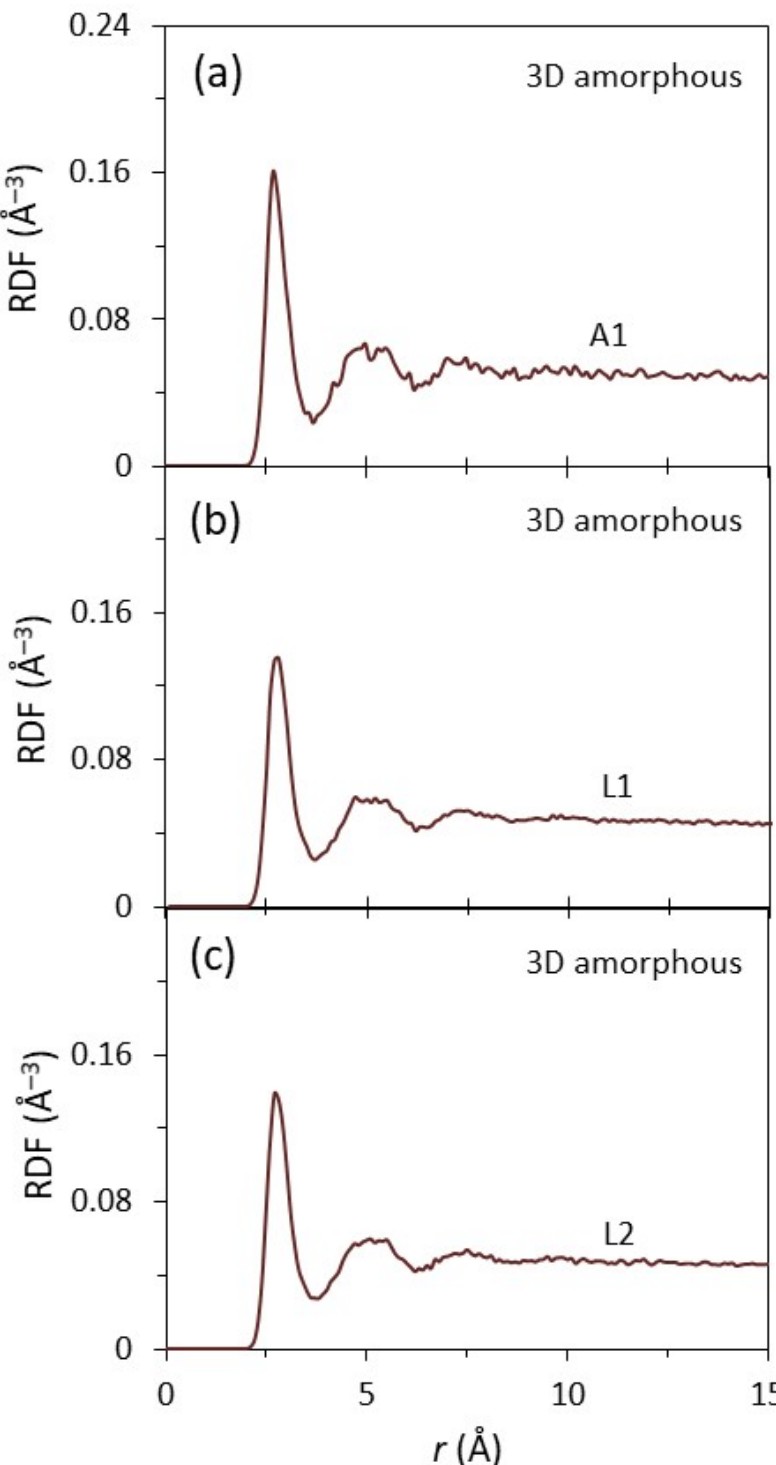

**Figure 4.** The radial distribution functions (RDFs) as a function of the distance, *r*, for (**a**) the A1, (**b**) L1, and (**c**) L2 of the liquid at the interface with 3D amorphous substrate equilibrated at *T* = 900 K. The 1st peak is sharp and the 2nd peak is diffuse in the curve of the RDFs for all the A1, L1, and L2, suggesting that there are only short-range order in the amorphous substrate with rough surface and the liquid at the interface.

Figure 5 shows the time-averaged atomic positions of the amorphous surface layer (A1), the first liquid Al layer (L1) superimposed on those of A1 (L1/A1), and the second liquid layer (L2) in the system with the 2D amorphous substrate at *t* = 1000 ps during the simulation equilibrated at *T* = 900 K. In general, A1, L1, and L2 were all disordered. The atoms in

A1 were fixed during the simulation. However, there appeared to exist some local ordered clusters in L1 (enclosed by a dashed square), which we discuss in more detail later.

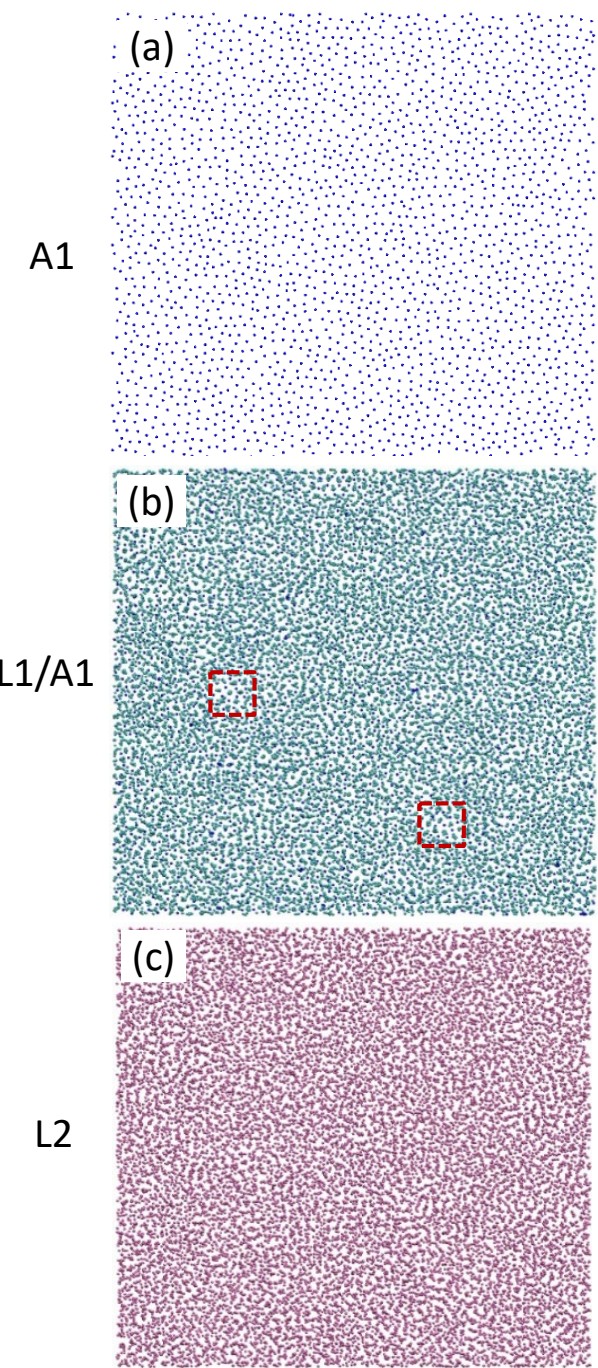

**Figure 5.** Time-averaged atomic positions of the (**a**) A1, (**b**) L1 on A1, and (**c**) L2 of the system with 2D amorphous substrate equilibrated at $T$ = 900 K. The A1 layer has a disordered structure. There are some local ordered structures (enclosed by the dashed squares) in L1 although both L1 and L2 are overall disordered.

### 3.2. Prenucleation and Heterogeneous Nucleation on Smooth Amorphous Surface

Nucleation was determined by monitoring the total energy, $E_t$, as a function of time, $t$, during the simulations, where $E_t$ includes the potential energy and vibration energy of all the atoms in the simulation system. We found that nucleation occurred at $T_n$ = 579 K for the system with a 2D amorphous substrate. Figure 6a shows the $E_t$ as a function of the

rescaled time, $t$, for the system during the simulation at $T_n$ = 579 K. $E_t$ starts to decrease from $t_1$ ($t$ = 0 ps), and levels off at $t_2$ = 290 ps, where $t_1$ and $t_2$ mark the start of nucleation and the end of the solidification, respectively, in the simulation system.

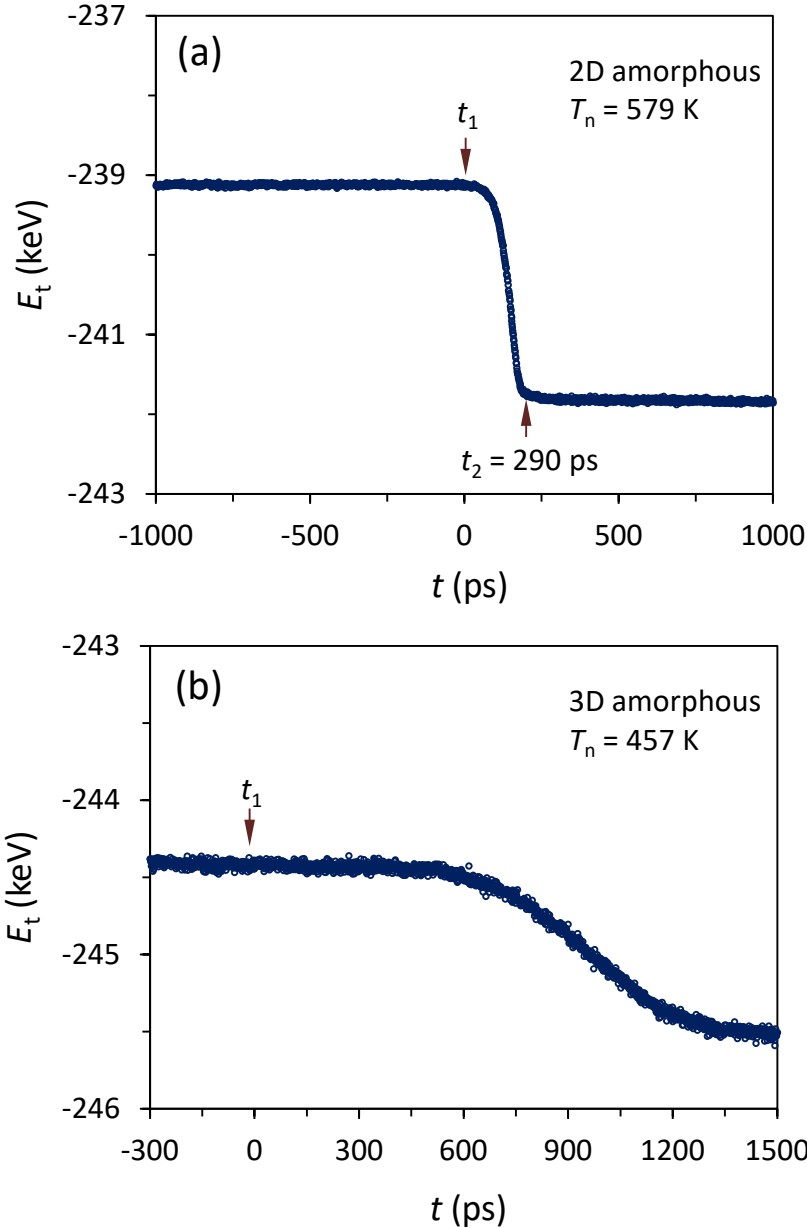

**Figure 6.** Total energy, $E_t$, as a function of the rescaled time, $t$, during the simulation at (**a**) $T_n$ = 579 K for 2D amorphous and (**b**) at 457 K for 3D amorphous systems. $E_t$ starts to decrease from $t_1$. This suggests that the nucleation starts at $t_1$, and the solidification of the system ends at $t_2$. The $T_n$ for the system with a smooth-surfaced amorphous substrate is more than 100 K larger than that for the system with a rough surface.

Figure 7 shows the time-averaged atomic positions of the L1/A1, L2, and L3 of the system with 2D amorphous substrate during the simulation at $T_n$ = 579 K. Before $t_1$ ($t$ = 0 ps), there are some locally ordered structures in L1 and L2, highlighted by the envelopes (Figure 7a,b). The size of these local ordered structures (Figure 7a,b) is substantially larger than those observed at temperatures above the liquidus (Figure 5), but still smaller than the critical size for a nucleus ($r^*$ = 0.9 nm, calculated with the homogeneous classical nucleation theory (CNT) [27]). These ordered clusters at $T_n$ are similar to the 2D ordered structures in the prenucleation observed in our previous simulations [3]. Thus, prenucleation also

occurs in a liquid at the interface with a smooth-surfaced amorphous substrate before the onset of nucleation.

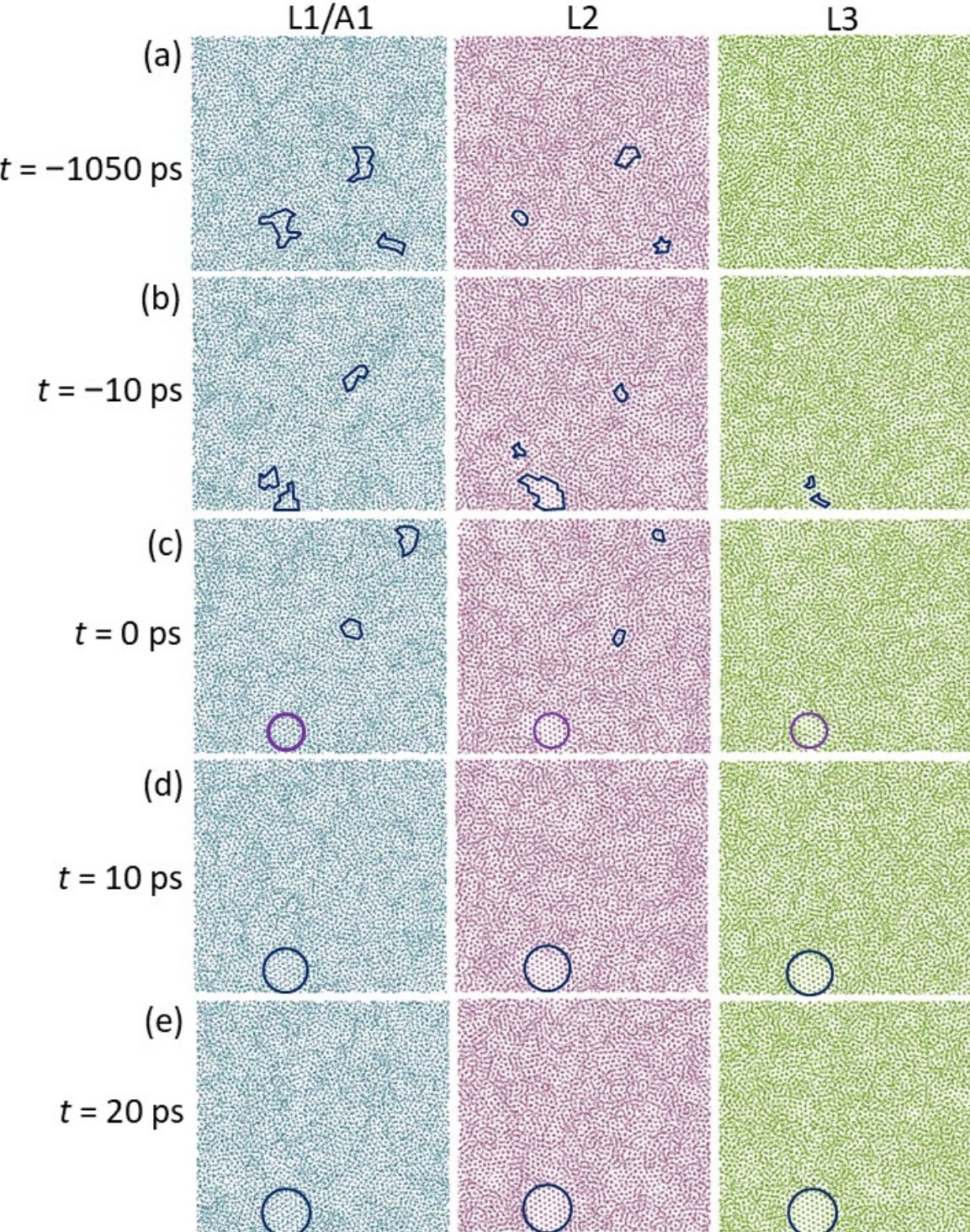

**Figure 7.** Time-averaged atomic positions in the L1/A1, L2, and L3 of the system with 2D amorphous substrate at (**a**) $t = -1400$ ps, (**b**) $-10$ ps, (**c**) 0 ps, (**d**) 10 ps, and (**e**) 20 ps during the simulation at $T_n = 579$ K. There are some solid clusters at the interface (enclosed by envelops) before $t_1$, and the nucleation starts by merging the solid clusters at $t_1$, producing a 2D nucleus in the L3 (highlighted by purple circles). The nucleus continues to grow in 3 dimensions from $t_1$ (highlighted by blue circles).

Nucleation proceeded by combining the 2D ordered structures into a solid cluster at the interface at $t = 0$ ps during the simulation at $T_n = 579$ K, which reached a critical size of $r^* = 0.9$ nm, as highlighted by the purple solid circles in Figure 7c. As a consequence, a 2D nucleus formed in L3, with a nearly perfect crystalline {111} plane of Al fcc structure. The nucleus continued to grow in three dimensions with time on the smooth surface of the amorphous substrate (Figure 7d,e).

### 3.3. Homogeneous Nucleation on Rough Surface

Nucleation occurred at $T_n = 457$ K for the system with a 3D amorphous substrate during the annealing with a temperature step of 1 K. Figure 6b shows the total energy ($E_t$) of the system as a function of rescaled time ($t$) during the simulation at $T_n = 457$ K. $E_t$ started to decrease from $t = 0$ ps ($t_1$), suggesting that the nucleation occurred at $t_1$ during the simulation at $T_n = 457$ K in the system of the liquid Al/amorphous substrate with rough surface.

Figure 8 shows the time-averaged atomic positions in the 20th (L20), 24th (L24), and 28th layer (L28) from the interface of the system with a 3D amorphous substrate during the simulation at $T_n = 457$ K, where L20, L24, and L28 represent the atomic layers in the liquid Al away from the interface with the amorphous substrate. The total thickness of the liquid was 36 atomic layers (8.42 nm) from the bottom to the top interfaces with the amorphous substrate. There was only a disordered structure in the liquid at $t = -200$ ps (Figure 8a). The cluster with ordered structure became visible in L24, at $t = 0$ ps (Figure 8b), where the solid cluster was enclosed by purple envelopes. The solid cluster reached a critical size of $r^* = 0.65$ nm (calculated with the homogeneous CNT [27] at an undercooling of 433 K), and then became the nucleus, which could continue to isothermally grow at the nucleation temperature (Figure 8c,d). L20 and L28 were still disordered. The nucleus continued to grow, and extended to L20 to L28 with a thickness of nine atomic layers (about 2.1 nm) at $t = 200$ ps. Therefore, the nucleation occurred inside the bulk liquid, being completely independent of the amorphous substrate surface. This confirms that the nucleation is homogeneous.

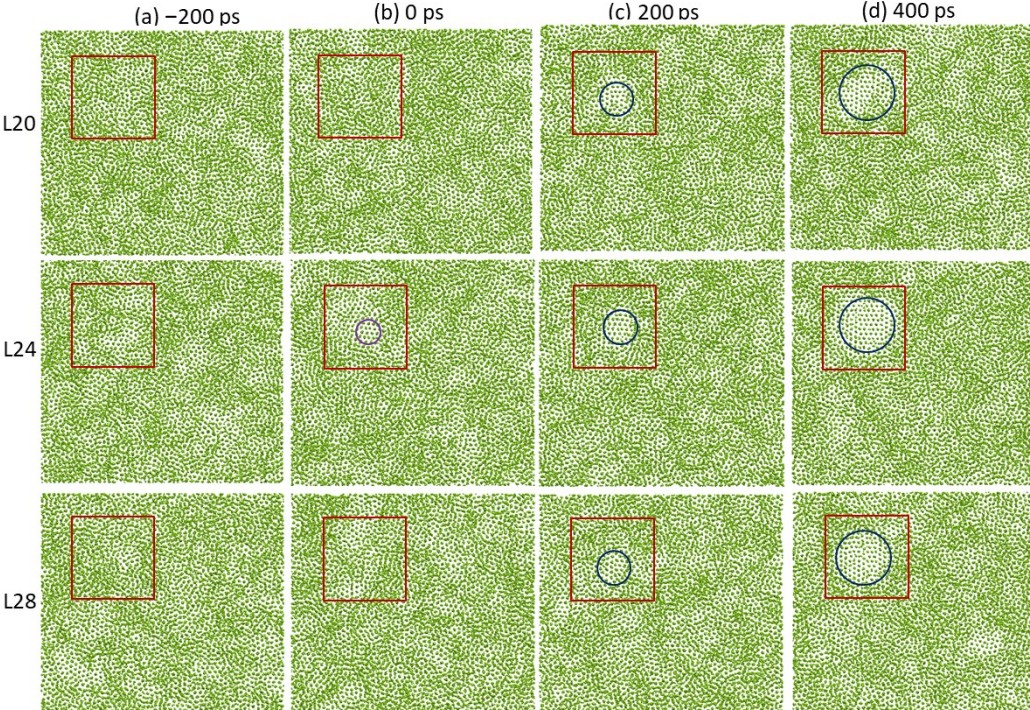

**Figure 8.** Time-averaged atomic positions of L20, L24, and L28 (away from the liquid/substrate interface) for the system with 3D amorphous substrate at a rescaled time (**a**) $t = -200$ ps, (**b**) 0 ps,

(**c**) 200 ps, and (**d**) 400 ps during the simulation at $T_n$ = 457 K. The red squares are used to mark the regions where the homogeneous nucleation occurs. There is a disordered structure at $t = -200$ ps, and a solid cluster reaches the critical size of $r^* = 0.65$ nm at 0 ps (highlighted by the purple circle). The nucleus then continues to grow with time (highlighted by blue circles).

## 4. Discussion

In reality, the occurrence of homogeneous nucleation of a solid is very rare in a liquid due to the inevitable existence of impurities, which are usually unknown in most cases. Impurities trigger heterogeneous nucleation at a relatively small undercooling compared with that of homogeneous nucleation. This study revealed that an amorphous substrate with a smooth surface can trigger heterogeneous nucleation, with an undercooling 100 K smaller than that required for homogeneous nucleation. Although the smooth surface of the amorphous substrate was artificially created in this study, it could be its natural status because the surface of an as-prepared amorphous substrate is usually atomically smooth due to its large surface tension [28]. Thus, the atomic layering can be induced by the smooth surface of an amorphous substrate, in which the clusters with similar structure to the stable phase are promoted by the atomic layering. Under extreme conditions for nucleation, such as large undercooling, heterogeneous nucleation may be facilitated by such atomic ordering at an undercooling slightly less than that of homogeneous nucleation. Consequently, this raises concerns regarding the accuracy of the measurement of the nucleation rate with the fluxing droplet method [21], in which the liquid droplet is enclosed inside the fluxing materials, which usually has a disordered structure with a smooth surface adjacent to the liquid.

This study further revealed that the structural templating is a universal atomistic mechanism for heterogeneous nucleation. The potency of a substrate strongly depends on the structural matching between the new phase and substrate, and degrades with increasing lattice misfit [29]. The lattice of a crystalline substrate can always provide a certain matching for the formation of the new phase as long as it is wetted by the liquid. For example, the coincidence site lattice (CSL) interface with good matching can form at the interface for a large lattice misfit, and may dramatically improve the potency of the original substrate [30]. Additionally, an amorphous substrate statistically has various small regions analogous to a variety of the crystal structures. It induces local ordered structures in some small regions in the liquid at the interface, which has a certain structural matching between the substrate and the stable phase. This structural matching promotes the occurrence of heterogeneous nucleation.

In this study, we bridged heterogeneous nucleation with homogeneous nucleation with decreasing potency of the substrate. Heterogeneous nucleation occurs as long as the substrate can provide structural templating; otherwise, homogeneous nucleation takes place. Without structural templating, it is not possible for the substrate to trigger heterogeneous nucleation. Instead, homogeneous nucleation is preferred through a fluctuation mechanism.

## 5. Summary

In this study, we investigated prenucleation and nucleation in liquid Al adjacent to amorphous substrates with smooth or rough surfaces. It was revealed that there was atomic layering in the liquid at the interface within about five atomic layers in the liquid at the interface with the smooth amorphous substrate surface, but without visible in-plane atomic ordering. There was neither atomic layering nor in-plane atomic ordering in the liquid at the interface with a rough-surfaced amorphous substrate. The amorphous substrate with a smooth surface could induce some local ordered structure in the liquid at the interface. At a nucleation temperature of 579 K, heterogeneous nucleation occurred in the liquid at the interface with smooth amorphous substrate surface, facilitated by the local ordered structure due to structural templating. A 2D nucleus was created in the third atomic layer at the end of the heterogeneous nucleation process. The amorphous

substrate with a rough surface could not trigger heterogeneous nucleation, and, in this case, homogeneous nucleation occurred at a nucleation temperature of 457 K. This study confirms that structural templating is a general mechanism for heterogeneous nucleation.

**Author Contributions:** H.M. conducted MD simulations, visualization, and original draft writing; Z.F. conducted conceptualization of the research, funding acquisition and supervision, and all the authors contributed to review and editing of the manuscript. All authors have read and agreed to the published version of the manuscript.

**Funding:** This work was funded by the EPSRC of the UKRI under grant number EP/N007638/1.

**Data Availability Statement:** All data are available in the main text.

**Conflicts of Interest:** The authors declare no conflict of interest.

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
