# Peer review of "Molecular Dynamics Simulations on Effect of Surface Roughness of Amorphous Substrate on Nucleation in Liquid Al"

_metals, doi:10.3390/met12091529_

Round 1

Reviewer 1 Report

The paper presents original and interesting research work. Here are my comments which should improve the reception:

1. In methods, please specify the parameters of the discussed surface. Is the structure of the rough surface irregular (random?), or deterministic (certain patterns are used). How do you model that rough surface? There is no information about the roughness while it is a crucial part of your research. The effect of surface texture on the physical or chemical interactions are dependent on the size and shape of the geometric features. These interactions manifest itself at the particular scale or scales of interaction. Please see the following paper for more information (section 3.9 to 3.11 in particular):

https://doi.org/10.1016/j.cirp.2018.06.001

2. Figure 1 is unclear. x and y scale should be given. What are the blue lines (a) and blue region in (b)? A more detailed description should be given.

3. Quality of Figure 2 should be improved.

4. It would beneficial to analyze different example of rough surfaces. This rough surface is vaguely described here what makes the results hard to generalize and provide insightful conclusion. I do agree with the authors that the roughness is an important factor in nucleation but there is no sound hypothesis or explanation why it is so. Heat transfer is also different for rough vs smooth surface. This should be better addressed in the paper.

Author Response

Authors’ Reply to Reviewers Comments

We would like to thank all the Reviewers for their effort to review our manuscript, and for their constructive and positive comments. Here are our rely to the comments and suggestion raised by the reviewers.

Reply to Reviewer 1

  • “In methods, please specify the parameters of the discussed surface. Is the structure of the rough surface irregular (random?), or deterministic (certain patterns are used). How do you model that rough surface? There is no information about the roughness while it is a crucial part of your research. The effect of surface texture on the physical or chemical interactions are dependent on the size and shape of the geometric features. These interactions manifest itself at the particular scale or scales of interaction. Please see the following paper for more information (section 3.9 to 3.11 in particular): https://doi.org/10.1016/j.cirp.2018.06.001

 Response: The amorphous substrate with rough surface is constructed from bulk amorphous, characterized by disordered structure of the bulk amorphous phase either in the bulk or at the surface. The roughness of the surface is at an atomic level, with the character of bulk amorphous phase, but can’t be called random, and the substrate surface has no particular patterns, and we could not discuss any further interactions. This Reviewer’s concerns here are originated from the confusion of atomic level surface roughness (in this paper) and engineering surface roughness (e.g., studied by https://doi.org/10.1016/j.cirp.2018.06.001). Atomically rough surface, as defined in Ref 17, describes the overall deviation of atomic positions relative to a perfect atomic plane, and such deviation is less than one atomic spacing. However, engineering surface roughness deals with surface topology at micro- or even macro-scales. An engineering rough surface may be perfectly smooth at atomic level.

  • “Figure 1 is unclear. x and y scale should be given. What are the blue lines (a) and blue region in (b)? A more detailed description should be given.”

 Response: The blue lines in Fig. 1(a) and blue region in Fig. 1(b) are the amorphous substrate with smooth surface and rough surface, respectively. They are front views of the time-averaged-atomic-positions. We had added explanations for the liquid Al and amorphous substrates in the revised text and in the caption of Fig. 1 for better understanding. For atomistic simulations, it usually characterizes the dimension of the simulation system with the number of atomic planes, as we stated in Section 2: Simulation approach. We have added dimensions of the simulation system in the revised text.

  • “Quality of Figure 2 should be improved.”

 Response: We have provided high-quality figures in the revised manuscript.

  • “It would beneficial to analyze different example of rough surfaces. This rough surface is vaguely described here what makes the results hard to generalize and provide insightful conclusion. I do agree with the authors that the roughness is an important factor in nucleation but there is no sound hypothesis or explanation why it is so. Heat transfer is also different for rough vs smooth surface. This should be better addressed in the paper.”

 Response: We agree with the reviewer 1’s suggestion that it would be helpful to generalize the result by including more systems with different surface roughness. This has been done in Ref 17, where we have systematically investigated the effect of atomically rough substrate surface on prenucleation. In this paper, we intend to demonstrate that the structure templating is a universal mechanism for heterogeneous nucleation, and then bridging heterogeneous nucleation (structural templating with smooth surface) with homogeneous nucleation (no structural templating with rough surface). For MD simulations, there is no problem of heat release accumulating at the interface, where the velocity of the atoms is simply rescaled to a certain temperature and then to achieve the equilibration of the system with the ensemble.

Reviewer 2 Report

The submitted manuscript is entitled “A Molecular Dynamics Study of the Effect of Surface Rough-ness of Amorphous Substrate on Nucleation in Liquid Al.”  

The authors use molecular dynamics (MD) simulations to investigate the atomic ordering and nucleation in the liquid aluminum (Al) adjacent to the amorphous substrate with a smooth and rough surface. The study reveals that structural templating is the general mechanism for heterogeneous nucleation in the present case.  The presented work seems to be sound, and the results appear to be of interest to readers of the journal and represent a scientific contribution to the field. 

This is the list of some minor problems those need to be repaired:

The total energy, Et, should be clearly defined in the manuscript.

The manuscript needs to be edited properly. All the equations should be presented within the text column. 

Figure 2 (a): the symbol ¶ should be removed.

Figure 4: the figures should be labeled as (a), (b), (c).

You need to read the manuscript carefully and check typos.

Author Response

We would like to thank all the Reviewers for their effort to review our manuscript, and for their constructive and positive comments. Here are our rely to the comments and suggestion raised by the reviewers.

Reply to Reviewer 2

  • “The total energy, Et, should be clearly defined in the manuscript.”

 Response: The total energy, Et, includes the potential energy and vibration energy of all the atoms in the simulation system. This has been added in the revised text.

  • “The manuscript needs to be edited properly. All the equations should be presented within the text column.” 

 Response: We have properly edited the manuscript, including all, the equations.

  • “Figure 2 (a): the symbol ¶ should be removed

 Response: The symbol ¶ in Figure 2 (a) has been removed.

  • “Figure 4: the figures should be labeled as (a), (b), (c).”

 Response: We have added the labels (a)-(c) in Figure 4.

Reviewer 3 Report

This paper is an interesting molecular dynamics study of heterogeneous nucleation of smooth and rough surfaces of crystalline or amorphous substrates. Very interesting is the result, that a smooth amorphous substrate can lead to local ordering and initiate thereby a layering that leads to heterogeneous nucleation. This is an effect, which was thought to be not possible, since a certain fit between the growing solid on a substrate was thought to be necessary for heterogeneous nucleation. The paper is an excellent numerical study, well performed with state-of-the art method of MD simulations. It cites the important literature and discusses the observations very  carefully. The scientific content is new and will induce discussions in the scientific community.   Therefore: the paper should be published as it is!

Author Response

We would like to thank Reviewer 3 for his/her effort to review our manuscript, and for his/her constructive and positive comments.

Reviewer 4 Report

Review of the work:

 A Molecular Dynamics Study of the Effect of Surface Rough-2 ness of Amorphous Substrate on Nucleation in Liquid Al

In this article, the authors investigate atomic ordering and nucleation in liquid aluminum (Al) adjacent to an amorphous substrate with either a smooth or a rough surface, using molecular dynamics (MD) simulations. At the interface with a smooth surface, the calculations indicate layering in the liquid, which exhibits approximately five atomic layers, but with no visible in-plane atomic ordering. However, the smooth surface of the amorphous substrate induces some locally ordered structure in the liquid at the interface through a structural template mechanism, which promotes heterogeneous nucleation by creating a two-dimensional (2D) nucleus in the third layer. On the other hand, the results show that no layering nor in-plane atomic ordering is present with the amorphous substrate with a rough surface. The calculation indicates this substrate has no effect on nucleation in the liquid, leading to homogeneous nucleation occurring with 100 K greater undercooling than heterogeneous nucleation on a smooth amorphous substrate. It is an excellent and relevant work that proposes a new explanation of nucleation during solidification, clarifying aspects that remained hidden until now.

I consider this an excellent work that can be of great interest to Metals readers and that it must be published after addressing the following minor corrections.

C1.-Please specify in the text for figures 1(a) and (b) that the substrate is shown at the bottom of both figures as horizontal lines for the smooth surface and a green atomic stripe for the rough surface.

C2.-In Figure 6(a), t1 and t2 are shown as the start and end times of solidification, while in Figure 6(b), only t1 is shown, although t2 would be around 1200 ps. Please include t2 in Fig. 6(b) and comment on observed differences between t1 and t2.

C3.-Please correct the following typos:

Figure 6 caption, page 9 line 268

Says: K for 2D amorphous and (d) 457 K …

Must say: K for 2D amorphous and (b) 457 K …

Page 8, title to the right of figure 5(b)

says: 1/A1

Must say: L1/A1

Page 12, line 355

Says: occur al long as the substrate…

Must say: occur as long as the substrate…

Best regards

Author Response

We would like to thank all the Reviewers for their effort to review our manuscript, and for their constructive and positive comments. Here are our rely to the comments and suggestion raised by the reviewers.

  • Please specify in the text for figures 1(a) and (b) that the substrate is shown at the bottom of both figures as horizontal lines for the smooth surface and a green atomic stripe for the rough surface.”

 Response: We have added the explanation for the liquid and amorphous substrates in Figure 1 and in the revised manuscript.

  • In Figure 6(a), t1 and t2 are shown as the start and end times of solidification, while in Figure 6(b), only t1 is shown, although t2 would be around 1200 ps. Please include t2 in Fig. 6(b) and comment on observed differences between t1 and t2.”

Response: It is very difficult to specify t2 (end of solidification) for the simulation with rough surface, but definitely later than t = 1200 ps. This is the case for occurrence of homogeneous nucleation in Figure 6(b), and t2 may not reach until the end of the simulation at Tn = 457 K. On the other hand, in Figure 6(a) the nucleation is heterogeneous.

  • “Please correct the following typos: Figure 6 caption, page 9 line 268, Says: K for 2D amorphous and (d) 457 K …, Must say: K for 2D amorphous and (b) 457 K …Page 8, title to the right of figure 5(b), says: 1/A1, Must say: L1/A1, Page 12, line 355, Says: occur al long as the substrate…, Must say: occur as long as the substrate…”

 Response: We have corrected the typos in the revised manuscript.

Reviewer 5 Report

1. What is the significance of B2O3 , as explained in the last paragraph of the introduction? 

 2. How possible to choose the predicted melting temperature for pure Al is 870 ± 4 K ? Better to take 660 C ...not that reference. 3. In figure 1, leveling should be done to know the interfacial area. 4. As per the figure and description, there are local ordering of atom observed, but not w.r.t the interfacial area. According to the objective is is devoting much. Justification is highly essential with modifications.  5. References to be formatted.

Author Response

We would like to thank all the Reviewers for their effort to review our manuscript, and for their constructive and positive comments. Here are our rely to the comments and suggestion raised by the reviewers.

(1). “What is the significance of B2O3, as explained in the last paragraph of the introduction?” 

 Response: For measuring nucleation undercooling and nucleation rate in metallic systems, B2O3 is often used to separate the droplets with very small size, and simultaneously to remove the impurity in the droplet as much as possible. Thus, it claimed that the homogeneous nucleation will occur in the smaller droplets since there may exist no impurity inside these small droplets. 

(2). “How possible to choose the predicted melting temperature for pure Al is 870 ± 4 K? Better to take 660 C ...not that reference.”

 Response: For MD simulations, the melting point of pure Al depends on the potential that used in the simulation, and the calculated melting point in the simulation may be different from the experimental value. The purpose of the MD simulation is to catch the fundamental information for the problem of concern, for instance, determination of nucleation undercooling, where the melting temperature of Al by MD (870K) should be used instead of the actual melting point for Al (933K).

(3). “In figure 1, leveling should be done to know the interfacial area.”

 Response: We intend to show the degree of the atomic layering at the interface of the liquid/amorphous substrate with smooth or rough surface in Fig. 1. An added line at the interface may blur the layering.

(4) “As per the figure and description, there are local ordering of atom observed, but not w.r.t the interfacial area. According to the objective is is devoting much. Justification is highly essential with modifications.” 

Response: There are some local orderings in the 1st liquid interfacial layer at T = 900 K, as highlighted by the red dashed squares (Fig. 5b). In the locally ordered regions, the atoms are localized in the time-averaged atomic positions (i.e., solid-like).

(5). “References to be formatted.” 

 Response: We have reformatted the references in the revised manuscript.

Round 2

Reviewer 1 Report

Ok

Reviewer 5 Report

The manuscript titled "A Molecular Dynamics Study of the Effect of Surface Rough- 2 ness of Amorphous Substrate on Nucleation in Liquid Al" has been revised significantly and can be accepted for the publication.